# Unraveling the Oncogenic Potential of VAV1 in Human Cancer: Lessons from Mouse Models

**DOI:** 10.3390/cells12091276

**Published:** 2023-04-27

**Authors:** Batel Shalom, Yaser Salaymeh, Matan Risling, Shulamit Katzav

**Affiliations:** 1Department of Developmental Biology and Cancer Research, Institute for Medical Research Israel-Canada, Faculty of Medicine, Hebrew University, Jerusalem 91120, Israel; 2Department of Military Medicine and “Tzameret”, Faculty of Medicine, Hebrew University, Jerusalem 91120, Israel; 3Medical Corps, Israel Defense Forces, Tel-Hashomer 02149, Israel

**Keywords:** VAV1, GEMMs, KRAS, p53, RAC1

## Abstract

VAV1 is a hematopoietic signal transducer that possesses a GDP/GTP nucleotide exchange factor (GEF) that is tightly regulated by tyrosine phosphorylation, along with adapter protein domains, such as SH2 and SH3. Research on *VAV1* has advanced over the years since its discovery as an in vitro activated oncogene in an NIH3T3 screen for oncogenes. Although the oncogenic form of *VAV1* first identified in the screen has not been detected in human clinical tumors, its wild-type and mutant forms have been implicated in mammalian malignancies of various tissue origins, as well as those of the hematopoietic system. This review article addresses the activity of human *VAV1* as an overexpressed or mutated gene and also describes the differences in the distribution of *VAV1* mutations in the hematopoietic system and in other tissues. The knowledge accumulated thus far from GEMMs expressing *VAV1* is described, with the conclusion that GEMMs of both wild-type *VAV1* and mutant *VAV1* do not form tumors, yet these will be generated when additional molecular insults, such as loss of p53 or KRAS mutation, occur.

## 1. Introduction

VAV1 is a hematopoietic signal transducer [1,2]. It was first identified in 1989 as an in vitro activated oncogene [1]. Hence, following transfection into NIH3T3 cells, the amino-terminus of VAV1 was deleted and replaced by sequences of pSV2neo, which was used as a selectable marker in the nude mouse tumorigenicity assay [1]. Since it was the sixth oncogene discovered in Barbacid’s lab by an Israeli scientist (S.K.), it was given the name Vav, which is derived from the sixth letter of the Hebrew alphabet [1].

VAV1 functions as a guanine nucleotide exchange factor (GEFs) for the RHO/RAC family of small GTPases [3], an activity controlled by tyrosine phosphorylation that leads to cytoskeleton reorganization and the control of various cellular functions. Although it is estimated that there are over 50 known human GEFs, only a few GEFs are tyrosine phosphorylation dependent, including VAV1 [3], Sos1 [4], C3G [5], and others [6]. VAV1 also mediates additional functions that are considered GEF independent, such as calcium release, stimulation of transcription factors such as Nuclear Factor of Activated T cells (NFAT), Activator Protein-1 (AP-1), and Nuclear Factor kB (NF-kB), and activation of ERK/MAPK [7,8]. Obviously, the vast array of VAV1’s functions leads to the regulation of cell proliferation, survival, and migration, thus placing it as a central protein in the hematopoietic system [7,8].

We review herein the current knowledge about the various aberrations of VAV1 in human cancer, which suggests that it can be considered an oncogene. We also focus on what we have learnt on the role of *VAV1* in cancer using genetically modified mouse models (GEMMs).

## 2. Structure/Function of VAV1

The VAV1 protein is endowed with multiple structural regions that are important for its function as a key signaling protein. Numerous reviews have described the domain structure of VAV1 and the function attributed to each domain [8,9,10,11]. Here, we will summarize briefly the current knowledge with relation to each domain’s function (Figure 1).

The VAV1 domains include: CH domain—a calponin-homology domain (amino acids 3–121) that enables VAV1 to control calcium mobilization [8,12]. Due to its interaction with the C1 region, it also participates in the regulation of the activity of the catalytic Dbl homology (DH) domain of VAV1 as a GEF [9]. Ac—An acidic motif (amino acids 133–193) that contains three regulatory tyrosines, Y142, Y160, and Y174, which need to be phosphorylated to facilitate the release of the autoinhibition of the DH domain, thus leading to enhanced VAV1 GEF activity towards RHO/RAC GTPases [10]. DH—a Dbl homology region (amino acids 199–373) that functions as a GEF towards the RHO/RAC GTPases [8,11,13] once VAV1 is activated by tyrosine phosphorylation [3]. PH—a Pleckstrin homology domain (amino acids 404–505) that potentially binds to PI5P and possibly other mono-PIs [12,14]. C1—an atypical C1 region (amino acids 515–564) that is involved in protein–protein interactions [15]. Structural studies demonstrated that the PH and C1 domains stabilize the DH domain structure and consequently contribute to efficient GEF activity [16]. Proline rich region (amino acids 606–610)—this enables the association of VAV1 to various SRC homology 3 (SH3) containing proteins [17]. SH2—an SRC homology 2 region (amino acids 672–746) that enables the binding of VAV1 to tyrosine phosphorylated proteins [18,19]. SH3—two SH3 regions (NSH3- amino acids 615–659 and CSH3- amino acids 786–841) that mediate interactions with proline-rich domains [18]. Numerous proteins were detected as binding to these regions, especially to the CSH3 [8]. Mutations in tyrosine residues within the CSH3 region were shown to affect its GEF activity, thus suggesting that it participates in intramolecular autoinhibition of the protein [20]. The CSH3 domain was also demonstrated to assist in E3 ubiquitin ligase Casitas B-lineage lymphoma B (CBL-B)-dependent ubiquitinoylation and proteasomal degradation of the intracellular fragment of Notch1 (ICN1) [21]. NLS—two nuclear localization signals (amino acids 487–494 and 576–589) [7].

Overall, VAV1’s domains play crucial roles in its ability to interact with other proteins and modulate signaling pathways. 

## 3. Biological Functions of VAV1

VAV1, which is exclusively expressed in the hematopoietic system, participates in various cellular responses. Since the role of VAV1 in the hematopoietic system has been extensively reviewed [8,22,23,24], it is only briefly summarized here. Though many of VAV1’s activities are attributed to its function as a GEF for the RHO/RAC family of GTPases, i.e., GEF dependent, it is also involved in GEF-independent functions.

GEF-dependent functions—Once VAV1 is tyrosine phosphorylated, its activity as a GEF is enhanced [3]. RAC1 is a preferred substrate for VAV1 [25,26], and its activity on CDC42, RHOA, and RHOG is enhanced, though to a lower extent [27]. VAV1 is tyrosine phosphorylated in immune cells as a result of the activation of receptors, such as the T-cell receptor (TCR) [18,19], B-cell receptor (BCR) [28], FcRI [29], cytokine receptors [30], NK receptors [31], chemokine receptors [32], and integrins [33]. Depending on the specific hematopoietic cell type, these receptors’ activation of VAV1 has distinct effects via VAV1’s function as a GEF. By functioning as a regulator of cytoskeleton organization, VAV1 has been implicated in the development of the immunological synapse (IS) in T cells [34] and B cells [35], and exhibits cytoskeleton rearrangement and immune signaling, as shown in studies of VAV1-deficient mice [36,37]. VAV1 GEF activity is also essential for the stimulation of killing by NK cells [38]. It has been shown to be required for phagocytosis mediated by RAC-dependent complements [39], cell migration [40], and macrophage chemotaxis to CSF-1 [41]. In addition, VAV1 is involved in regulating cell migration and invasion, properties that are critical for tumor cell metastasis [42,43].

GEF-independent functions—These functions are primarily mediated by the association of VAV1 to other proteins through various adapter domains. For example, VAV1 plays a role in the activation of transcription factors such as NFAT, AP-1, and NF-kB [44,45] following TCR stimulation. It was demonstrated that VAV1’s capacity to initiate the release of calcium from T cells’ inner reservoirs is essential for this activity [45]. VAV1 can regulate apoptosis (programmed cell death) by modulating the activity of pro-apoptotic and anti-apoptotic signaling pathways [46]. VAV1 contributes to the DNA damage response by promoting cell survival and cell cycle progression in a GEF-independent manner [47]. VAV1 is also involved in the regulation of oxidative stress responses and serves as a scaffold protein to activate signaling pathways through protein–protein interactions [48]. In addition, VAV1 promotes extracellular signal-regulated kinase (ERK) and c-Jun N-terminal kinase (JNK) pathways [3,49,50]. It also initiates phospholipase Cγ1 stimulation following T cell receptor (TCR) activation and is involved in the recruitment of Son of Sevenless (Sos1) and Sos2 to the transmembrane adapter protein linker of activated T cells (LAT) [51]. 

In summary, VAV1 has been found to be a versatile signaling transducer essential for numerous biological activities in the hematopoietic system through its functions in GEF-dependent and GEF-independent pathways.

## 4. VAV1 in Human Cancer

*VAV1* was originally identified as an in vitro activated oncogene [1] rather than as a true oncogene. Though this revelation was very disappointing for the researchers, it clearly indicated that the gene must be physiologically important for normal cells since changes in its structure can lead to transformation. Indeed, several years later, when the wild-type *VAV1* was isolated and studied [52,53], the function of VAV1 as a signal transducer was understood [11,18,19], structural attributes controlling VAV1 function were revealed [54], and the mode of conversion of wild-type *VAV1* into an oncogene was fully evaluated. The analysis of the basis of regulation of its GEF activity provided an explanation as to why loss of the N-terminus leads to its activation as an oncogene [10]. Thus, the phosphorylation of Y174 (pY174) in the Ac domain abolishes the autoinhibition of the DH domain [10]. Therefore, N-terminal truncation or mutation of this residue leads to the stimulation of its GEF activity, a property required for its transforming activity [10]. Loss of the N-terminus sequences allows the protein to convert to an open form, which is required for its full activity as an uncontrollable GEF protein; thus, it is capable of transforming cells. The mystery was solved, but the following question remained: Is VAV1 a protein involved in human cancer?

Katzav’s laboratory was the first to report *VAV1* overexpression in human neuroblastoma specimens, suggesting that when VAV1 is expressed in tissues other than hematopoietic tissues, it may be involved in cancer [55]. *VAV1* mutations were recently reported in human cancers of various tissue origins (https://cancer.sanger.ac.uk/cosmic/). Data collected on VAV1 presence and involvement in human cancers, either as an overexpressed or mutated protein, suggest that it is an oncogene, as presented below.

## 5. *VAV1* Ectopic Expression in Human Tumors

After showing that VAV1 is overexpressed in SK-N-MC, a human neuroblastoma cell line, Hornstein et al., [55] showed its ectopic expression in 76% (32/42) human neuroblastoma specimens. Molecular analysis showed no gross rearrangements or mutations in the *VAV1* gene in SK-N-MC cells [55]. This study has led to intensive research in this area, yielding numerous publications, and to the recognition that *VAV1* might be involved in human tumorigenesis [56,57,58,59,60,61,62,63]. Several of these studies are detailed below.

VAV1 ectopic expression in human solid tumors—Billadeau’s group showed that VAV1 was expressed in 50% (48/95) of pancreatic ductal adenoma carcinoma (PDAC) samples [56]. VAV1 expression was also detected in 42% (33/78) of lung cancer cell lines and 44% (26/59) of primary human lung cancer tissue samples [57]. Furthermore, VAV1 expression correlated with estrogen receptor expression was noted in 42% (40/65) of primary human breast cancers [58]. Another study showed VAV1 expression in 96% (132/137) of invasive breast tumors from patients without lymph node involvement [59]. In most of these cases, VAV1, which is usually expressed in the cytoplasm, was localized to the nuclei of tumor cells [59]. VAV1 expression was also reported in 59% (52/88) of ovarian cancer [60], esophageal squamous cell carcinoma (ESCC) [61], human gastric cancer [63], and sonic hedgehog (SHH) subgroup medulloblastoma (MBSHH) tumors [62]. 

Most of the studies that reported an overexpression of VAV1 in human cancer also pointed to the correlation between VAV1 expression and tumor severity. Thus, positive-VAV1 PDACs [56], ovarian cancer [60], and MBSHH [62] had poorer survival rates compared to Vav-negative tumors. Several reports linked VAV1 expression and tumor size. For instance, high-intensity VAV1 expression was associated with larger lung cancer tumor size [57], MBSHH tumors [62], gastric cancer [63], and ESCC [61]. High levels of nuclear VAV1 were positively associated with low recurrence rates, regardless of breast tumor phenotype and molecular subtype [59].

VAV1 overexpression in hematopoietic malignancies—VAV1 has also been shown to be deregulated in hematologic malignancies [21,64,65,66]. Thus, it was detected as overexpressed in 13% of B-cell non-Hodgkin Lymphoma (B-NHL) and 34·4% of B-cell chronic lymphocytic leukemia (B-CLL) [64]. The majority of VAV1-positive tumors also displayed high levels of constitutive VAV1 tyrosine phosphorylation [64]. VAV1 overexpression was also associated with activation and higher proliferative activity of diffuse large B-cell lymphoma (DLBCL) [65]. VAV1 expression was also significantly higher in acute myeloid leukemia (AML) patients than in control tissues [66]. Those AML patients with high VAV1 expression were less likely to achieve complete remission compared to those patients with low VAV1 expression, thus suggesting that VAV1 expression levels have prognostic value [66]. 

Robles-Valero et al. proposed that VAV1 functions as a tumor suppressor gene in immature T cells and its loss leads to the development of T-cell acute lymphoblastic leukemia [21,67]. However, there are no reports supporting this classification of VAV1 in solid tumors of epithelial origin or other cancers. 

The methylation status of the *VAV1* promoter is a possible mechanism contributing to the ectopic expression of VAV1 in cancers of non-hematopoietic origin [56,62,68]. The *VAV1* promoter is completely unmethylated in human cells but methylated in cancer cells, resulting in its expression. Fernandez-Zapico, et al. [56] showed that epigenetic modifications rather than gene amplification led to VAV1 expression in pancreatic cancer cell lines. These results were also validated by Huang et al. [68], who further identified three *VAV1* gene bodies and promoter-CpG sites in PDAC patients involved in its ectopic expression in cancer via epigenetic mechanisms. Hypomethylation was also shown to cause ectopic expression of *VAV1* in MBSHH, confirming that VAV1 is an epigenetically regulated oncogene [62]. Transcription factors can also lead to the aberrant regulation of VAV1 expression [69]. Mutations in putative transcription factor binding sites in the *VAV1* promoter were shown to modify *VAV1* transcription in lung cancer cell lines [69].

What have we learnt about VAV1’s function in cancer when it is overexpressed? As discussed, Vav1 activates the RAC/RHO GTPases following tyrosine phosphorylation by various hematopoietic cell-surface receptors [18,19,28,29,30,31,32,33]. The hematopoietic cell-surface receptors do not encode tyrosine kinases, thus, tyrosine phosphorylation of VAV1 is carried out by cytoplasmic SRC family tyrosine kinases that are activated following the ligation of these receptors, such as LCK [70], FYN [71], HCK [72], and SYK [73]. VAV1 can be activated by various growth factor receptor tyrosine kinases (RTKs) that are usually expressed in tissues of different histologic origins, such as the receptors for EGF [18], PDGF [19], CSF1 [74], kit [74,75], and HGF [76]. Unlike the mode of its tyrosine phosphorylation following the activation of hematopoietic cell-surface receptors, RTKs that possess intrinsic tyrosine kinases phosphorylate VAV1 directly when expressed in the same cells [18,19,74,75,76]. Furthermore, VAV1 associates with RTKs such as EGFR and PDGFR through its SH2 domain [18,19]. Figure 2 depicts simplified representations of the complex signaling pathways initiated by TCR ligation (A) and EGFR activation (B), aiming to illustrate the basic concepts involved.

It is therefore not surprising that overexpression in various tissues leads to its tyrosine activation, thereby initiating many signaling pathways, including the activation of RHO/RAC GTPases. For example, VAV1 protein from SK-N-MC neuroblastoma cells was similar to wild-type VAV1 in terms of molecular size, phosphorylation state, and ability to associate with tyrosine-phosphorylated EGFR through its SH2 domain [55,77]. It has been shown that VAV1 requires both its GEF function leading to RAC1 activation and downstream signaling pathways to contribute to the oncogenicity of pancreatic cancer cells [56]. Interestingly, when VAV1 was deleted in VAV1-positive cells of the pancreas, these cells lost their ability to proliferate (in vitro) or carry out tumorigenesis (in vivo), even in the presence of mutant KRAS [56]. Similar results were obtained in human lung cancer cell lines, where the siRNA-mediated knockdown of VAV1 suppressed growth on agar and suppressed tumor growth in nude mice [57]. VAV1 is tyrosine phosphorylated in lung cancer cells upon activation by growth factors EGF and TGFα, suggesting its involvement in signaling pathways in non-hematopoietic tissues [57]. Using two different breast cancer cell lines, MCF-7 and AU565, Sebban et al. [78] demonstrated that AU565 cells expressing VAV1 have an increased expression of proliferation-related genes, whereas MCF-7 cells expressing VAV1 exhibited increased apoptosis-related genes [78]. Due to the differences in p53 expression in these cells, it was concluded that VAV1 may act as an oncogenic stress activator in cancer and its pro-apoptotic effects in mammary cells are p53 dependent [78]. The possibility that the overexpression of VAV1 may be involved in EMT in ovarian cancer was raised since morphological changes and the downregulation of E-cadherin expression, along with the upregulation of Snail and Slug, were observed in a human, high-grade, serous ovarian cancer cell line (SKOV3) [60].

VAV1 can also influence the tumor microenvironment in an autocrine/paracrine manner, potentially leading to tumor growth. Furthermore, VAV1 is involved in the release of cytokines and growth factors, such as TGFα [79] and EGF [80]. VAV1-dependent growth factor secretion was shown to occur in lung cancer cells that ectopically express the gene [58]. Colony-stimulating factor-1 (CSF1), a hematopoietic growth factor, is one of these growth factors [58]. The activation of the CSF1-R leads to VAV1 tyrosine phosphorylation and, consequently, to increased GEF activity [58]. In the H358 lung cancer cell line, CSF1 knockdown resulted in significantly smaller tumors that exhibited increased fibrosis and decreased tumor-infiltrating macrophages. Furthermore, VAV1 and CSF1 expression were positively correlated with tumor grade by immunohistochemical analysis of primary human lung cancers [58]. Thus, it has been suggested that lung cancer may involve interactions between cancer cells and the microenvironment regulated via the CSF1/VAV1 signaling pathway. The lessons we learnt from GEMMs about the involvement of VAV1 in shaping the tumor microenvironment are discussed below [81,82,83].

Another mechanism of action of VAV1 involvement emerged from studies in gastric tumors that showed that the ectopic expression of the RHOA mutant G17E in the human gastric cancer cell line MKN74 triggers VAV1 expression at the mRNA and protein levels, promoting cell migration and invasion [84]. RHOA G17E has been shown to promote cell invasion through VAV1 in vivo in a nude mouse peritoneal xenograft model [84]. This study raises the possibility that VAV1 acts on tumorigenesis through its association with the proteins of the RHO/RAC GTPase signaling pathway.

## 6. Mutations of VAV1 in Human Cancer

Since *VAV1* was detected, numerous studies have been conducted to untangle its structure/function by generating mutations at the DH, PH, C1, the SH3, and the SH2 domains of wild-type *VAV1* [9,20,85,86]. These experimentally introduced mutations did not result in increased transformation of NIH3T3 fibroblasts, apart from an artificially introduced mutation at the CSH3 region, amino acid residue 797 (D797N), which was transforming in vitro [87] and was also found in human cancer (http://cancer.sanger.ac.uk). Recent publications reported the numerous mutations found in *VAV1* in human tumors, either in solid tumors [88] (http://cancer.sanger.ac.uk) or in hematopoietic malignancies [89] (http://cancer.sanger.ac.uk) (Figure 3). 

### 6.1. VAV1 Mutants in Human Solid Tumors

The sequencing of human cancers of multiple tissue origins by the Human Genome Sequencing program (Wellcome Trust Sanger Institute) demonstrates that *VAV1* is mutated in ~1% of the tumors (http://cancer.sanger.ac.uk). Mutations in *VAV1* identified in human cancers involve all major domains (Figure 3A). The first study to report mutations in *VAV1* was based on the genomic analysis of non-small cell lung cancer (NSCLC) that included lung adenocarcinoma (660 samples) and lung squamous cell carcinoma samples (484) [88,89]. The *VAV1* mutations detected were of missense changes, internal deletions due to splicing defects, C-terminal truncations, and 3′-terminal translocations with other genes [88,89]. The activities as transforming genes of most of the mutants reported by Campbell et al. [88] and the Wellcome Trust Sanger Institute (http://cancer.sanger.ac.uk) have not yet been analyzed. E59K, D517E, and L801P, all VAV1 mutants detected in lung cancer, were studied by the Katzav laboratory and will be discussed below [90,91].

**Figure 3 cells-12-01276-f003:**
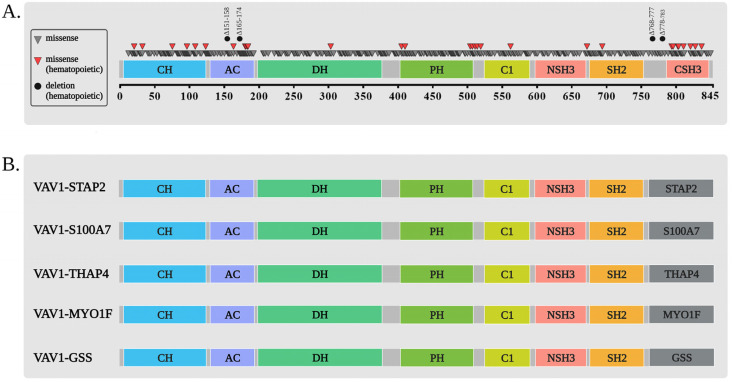
Summary of *VAV1* missense/deletion mutations (**A**) and VAV1 fusion proteins (**B**). (**A**): The localization of missense *VAV1* mutations (grey triangles) identified in cancers of various tissue origins are marked (https://cancer.sanger.ac.uk/cosmic/). Missense mutations (red triangles) and deletions (black dot) in Vav1 in hematopoietic malignancies are indicated (https://cancer.sanger.ac.uk/cosmic/). (**B**): Fusion proteins of VAV1 in which the CSH3 was replaced by other proteins identified in PTCL-NOS [67,92,93], AITL [89], ALCL [92], and ATL [94] are indicated. This figure was created with the BioRender.com program.

### 6.2. VAV1 Mutants in Human Hematological Malignancies

The first reports about rearrangements/mutations in *VAV1* in hematologic malignancies appeared in the last decade. *VAV1* gene alterations, including translocations and missense/deletion mutations, account for 18% of adult T-cell leukemia/lymphoma (ATLL), 11% of anaplastic large cell lymphoma (ALCL), 7% to 11% of peripheral T-cell lymphomas not otherwise specified (PTCL-NOS), and 5% in angioimmunoblastic T-cell lymphoma (AITL) (reviewed in [95]). The various molecular lesions in these malignancies are detailed below. Thus, VAV1 missense mutations and focal in-frame deletions are concentrated in the Ac region (E175K, E175V/L, D165_174, Y174C, L177R), the PH region (K404R, Q498K/R, M501R/L/N), the C1 region (E556D/K), the NSH3 (P615L), and the CSH3 (R790C, D797N/H, R798P/Q, K815E, R822Q/L) [67,89,92,93] (Figure 3A).

*VAV1* fusions, whose CSH3 is truncated and replaced by other genes, have been detected in various hematological malignancies. These events are often caused by an alternative splicing mechanism driven by focal deletion. Exon 25 of *VAV1* is misplaced into a potential intra-exonic splice acceptor motif of exon 26, thereby generating an mRNA with an in-frame deletion. Thus, *VAV1* fusions with GSS and MYO1F partners were detected in 11% of PTCL-NOS [92]. Additional *VAV1* fusion genes have been identified in PTCL and AITL, including *VAV1-THAP4*, *VAV1-MYO1F*, and *VAV1-S100A7* [67,89]. The fusion of *VAV1* to CD28 in PTCL has also been reported [96]. In *VAV1-S100A7*, the CSH3 domain of VAV1 was replaced with a full length S100A7, a calcium-binding epidermal protein with a proposed antibacterial and chemoattractant role [97]. The *VAV1-MYO1F* fusion is generated by fusion to the CSH3 domain of MYO1F, an actin-interacting motor protein. Finally, the VAV1-THAP4 fusion replaces the CSH3 domain of VAV1 with the C-terminal nitrobindin domain of THAP4 [98] (Figure 3B). 

VAV1 can also contribute to cancer in angioimmunoblastic T-cell lymphoma (AITL), with somatic RHOA^G17V^ mutations found in 50–70% of cases [99,100]. Upon TCR activation, VAV1 specifically binds to RHOA^G17V^, leading to the phosphorylation of VAV1 and the activation of the PLCγ1-SLP76-LAT signaling complex [89], thereby stimulating T cell proliferation and invasion and promoting AITL development.

### 6.3. Differences in Mutations in VAV1 between Solid and Hematopoietic Human Malignancies

Although mutations in *VAV1* in human solid tumors involve all of its major domains, the profile of *VAV1* mutations/alteration in human hematopoietic malignancies is distinctly different (Figure 3). It is noteworthy that the majority of mutations/changes in human hematopoietic malignancies are concentrated in the Ac and CSH3 domains, and there are only a few in the DH region. This is in sharp contrast to the *VAV1* alterations in all coding regions detected in solid tumors (Figure 3). 

There are other examples of genes with specific mutations that differ between hematopoietic and epithelial tissues. For example, certain mutations in the p53 gene, such as R248W, are associated with an increased risk of hematological cancers such as acute myeloid leukemia (AML) [101] and with chronic lymphocytic leukemia (CLL) [102]. On the other hand, other mutations in the p53 gene, such as R273H, are associated with an increased risk of epithelial cancers such as cutaneous squamous cell carcinoma [103]. Furthermore, the 185delAG mutation was associated with an increased risk of breast cancer in epithelial tissues, but not in hematopoietic tissues [104]. An additional mutation in the p53 gene, R273H, is associated with several types of hematopoietic cancers, including AML, chronic myelogenous leukemia (CML), and myelodysplastic syndrome (MDS) [105]. Differences in p53 mutations in cancer found in hematopoietic and epithelial tissues may be the result of the generation of hotspot mutation alleles that promote cancer in specific tissues [106]. Another example of mutational differences produced in cancers of different origins is the KRAS gene. Studies have shown that mutations in various domains of the KRAS protein can have tissue-specific effects. For example, the KRAS mutation G12C is associated with an increased risk of lung cancer, and G12D is associated with an increased risk of pancreatic cancer [107]. The different allele frequencies in different cancer types may be due to several factors, including the role of the KRAS gene in different signaling pathways and the intracellular environment in which mutations occur [107].

The cause for the apparent differences in the distribution of *VAV1* mutations among solid tumors compared to that in hematopoietic malignancies is not known.

### 6.4. Function of VAV1 Mutants in Human Cancer

Apart from the accumulation of information on the existence and location of many mutations detected in VAV1 in solid human tumors, there are still gaps in the biological understanding of the function of these mutants.

Shalom et al. [90] tested the oncogenicity of three mutations identified in human lung adenocarcinoma: E59K in the CH region, D517E in the C1 domain, and L801P in the CSH3 region [88]. E59K and D517E were found to be highly transforming in assays such as proliferation rate, growth on agar, and tumorigenesis in immune-compromised mice, whereas L801P was non-transforming [90]. The high transforming activity of these mutants was accompanied by increased GEF activity that stemmed from a truncated E59K protein and a highly stable overexpressed D517E protein [90]. Evidence for the altered biochemical behavior of VAV1 mutants was well explained by the computationally predicted properties we reported [91]. These results also emphasize the need to analyze different mutations biologically, since as shown by Shalom et al. [90], an existing mutation, such as L801P, does not necessarily show enhanced biochemical and transforming activities. Several of the mutations identified in lung cancer as well as hematological malignancies lead to increased VAV1 GEF activity, such as Y174C, E175K, E175V/L, L177R, K404R, Q498K/R, M501R/L/N, E556D/K, and P615L [108]. Other mutants, such as R790C, D797N/H, R798P/Q, K815E, and R822Q/L, show reduced/altered protein–protein interactions [105]. Additionally, there are examples of mutants such as D165_174, a deletion mutation that removes 10 amino acids within the DH domain of VAV1, resulting in a decrease in the GEF activity of VAV1, thus inhibiting cancer cell migration and invasion P615L [108]. Additional missense mutations analyzed, H337Y, E556D, R798P, E157K, K404R, Q498K, and M501R, showed weaker and more variable effects compared to wild-type VAV1, with only VAV1 Q498K showing clear increased ERK1/2 phosphorylation and modestly higher levels of phospho-PLCγ1 activation [67].

The contribution of the VAV1 fusion proteins *VAV1-MYO1F*, *VAV1-S100A7*, and *VAV1-THAP4*, generated by deletions of the CSH3 region, was confirmed in an experimental system in which the fusion genes were expressed in JURKAT J.VAV1 cells (Jurkat cells that lack the expression of wild-type VAV1) expressing VAV1 deleted at its CSH3 (Δ778–786) [67]. The results indicated that these fusion mutants lead to enhanced VAV1-dependent functions downstream of RAC1, including the increased phosphorylation of Y174, increased phosphorylation of ERK1/2, and increased JNK activation [67]. The same experimental approach also increased NFAT activity, a GEF-independent VAV1 activity that was further increased upon TCR stimulation [67].

The CSH3 domain of VAV1 occludes the access of VAV effector factors to the catalytic GEF domain by folding over to the CH and PH domains [109]. Therefore, it was postulated that a deletion in this region and fusion with other partners results in an open and active VAV1 organization, leading to increased GEF activity [67]. 

## 7. VAV1 and Genetically-Engineered Mouse Models (GEMMs)

The contribution of wild-type *VAV1* [81,82,83] or *VAV1* mutants [110,111] has recently been studied in vivo using GEMMs, as detailed below.

GEMMs expressing wild-typeVAV1— The expression of wild-type VAV1 in specific organs, such as the pancreas [81] or the lungs [83] indicated that although the expression of VAV1 alone did not lead to tumor development, the co-expression of VAV1 with mutant KRAS (KRAS^G12D^/VAV1 mice) synergistically enhanced tumorigenesis in these organs [81,83]. Moreover, of great interest is the fact that lung cancer development in KRAS^G12D^/VAV1 mice is accompanied by marked changes in the organization of the tumor microenvironment, particularly the appearance of tertiary lymphoid structures (TLSs) [83]. The expression of wild-type VAV1 in transgenic mice using the ubiquitous ROSA26 promoter resulted in VAV1 epithelial expression in various organs, including the pancreas, liver, and lung [82]. Surprisingly, no cancer developed in the epithelium of these organs, however, B-cell lymphomas developed [82]. The expression of CSF1 was elevated in the epithelial compartment of Rosa26 VAV1 mice, whereas the CSF1 receptor (CSF1R) was highly expressed in B-cell lymphomas, thus raising the possibility that CSF1 released from epithelial cells overexpressing VAV1 led to increased signaling in B-cells expressing CSF1R, leading to the development of B-cell lymphomas [82]. We also noted that although RAC1 activation was increased in the PDAC GEMM model [81], ERK phosphorylation, but not RAC activation, was increased in the lung [83] and B-cell lymphoma [82] mouse models. Our results summarized in Figure 4 provide novel insights as to VAV1′s contribution to tumor propagation and malignant progression.

GEMMs expressing mutant VAV1—Fukumoto et al. [110] established a GEMM model that ectopically expresses VAV1 gain-of-function mutations, a deletion mutant at the Ac region (165_174del), and the VAV1-STAP2 fusion gene under the CD2 promoter, which is expressed early in human thymocyte development and is found on about half of thymocytes and nearly all mature peripheral T cells. For up to a year, no tumors were observed in mice expressing the VAV1 mutants, but tumors developed when these mice were crossed with p53-null background mice [110]. In a different experimental system, an analysis of tumorigenesis in the immune system showed that in adoptive T cell transfer experiments, a trivalent subclass VAV1 mutant can directly induce the development of AITL in vivo [108]; this mutant is a VAV1 mutant allele that produces a CSH3-truncated protein (amino acids 835–845) and exhibits strong RAC and NFAT activity but is unable to inhibit ICN1 signaling. This pro-tumorigenic effect engages both the GEF-dependent and GEF-independent activities of VAV1 mutants [108]. In a gene-edited mouse model, Robles-Valero et al. [111] showed that the trivalent functional subclass of VAV1 mutations plays a critical role in the development of AITL when the suppressor TP53 is inactive. The requirement for the loss of TP53 may be related to problems such as overcoming the negative selection likely imposed on thymocytes expressing the active version of VAV1 [111]. Taken together, even severe mutations that deregulate and activate VAV1 function are insufficient to act alone as oncogenes and require additional molecular lesions. Furthermore, mice with a VAV1-CSH3 deletion mutation and loss of p53 did not develop tumors in the lung [111]. However, when crossed with mice expressing mutant KRAS in the lung, NSCLC tumors developed [111], clearly demonstrating that oncogenic mutants of VAV1 act synergistically with mutant KRAS to cause lung cancer (Figure 5). These results are in agreement with the reports from the Katzav laboratory that showed that even wild-type VAV1 is sufficient to co-operate with mutant KRAS in generating PDAC [81] and NSCLC [83].

## 8. Open Questions and Conclusions

The structure, function, and activity of VAV1 as a signal transducer in the hematopoietic system have all been extensively studied over the 30 years since its discovery. Although it was studied purely in vitro for a long period, the investigation of its role in cancer has finally shifted to studies in human specimens and GEMMs [81,82,83,110,111]. VAV1, either as an overexpressed protein or a mutant protein, is present in cancers originating from all tissue types, supporting its designation as a legitimate oncogene. However, several questions remain unanswered. For example, why are there distinct differences in *VAV1* mutations in hematopoietic malignancies compared to all other cancers? Do the mutations confer different growth and/or tumorigenic properties to tumors in different tissues? Furthermore, it is still unknown whether the majority of *VAV1* mutations found in human tumors function as bona fide oncogenic drivers in vivo, and if so, whether they function alone or together with other malignancies. Studies cited in this review demonstrate that although some mutants are highly transforming, others are not [90,91]. The number of downstream signaling pathways that need to be established is also unknown. The use of GEMMs has provided important clues regarding the role of *VAV1* in carcinogenesis in vivo. Tumors in GEMMs developed with either wild-type *VAV1* or mutant *VAV1* only when mice were crossed with either p53 [110,111] or mutant KRAS [81,83,111]. These results were obtained first by the Katzav lab using wild-type *VAV1* [81,83] and then by other labs using mutant *VAV1* [110,111]. The outcome of the studies using gene-edited mouse models [111] was similar to those using the overexpression of wild-type *VAV*1, driven by strong tissue-specific promoters [81,83], despite the fact that the studies using wild-type *VAV1* were criticized for using models that were somewhat artificial. Interestingly, the expression of VAV1 mutants or wild-type *VAV1* did not give rise to tumors, even after long periods of time since onset [81,82,83,110,111]; yet, whether there are *VAV1* mutants that could lead to tumor generation when expressed alone in GEMMs remains to be tested. Most knowledge about signaling pathways initiated by *VAV1* GEMMs comes from those using wild-type *VAV1* [81,82,83]. RAC1-GTP levels increased sharply in the KRAS^G12D^/VAV1 mice developing PDAC, whereas ERK phosphorylation remained at a level comparable to that of the pancreas of KRAS^G12D^ mice [81]. However, ERK phosphorylation was significantly increased in the lung tumors developed in KRAS^G12D^/VAV1 mice, whereas RAC1 activation remained constant [83]. These results suggest that different signaling pathways may be triggered by different tissues and microenvironmental signals.

Of particular interest is the *VAV1*-Rosa26 GEMM model in which B-cell lymphoma developed [82]. In this model, only VAV1 was expressed driven by the ubiquitous Rosa promoter. VAV1 was expressed in the epithelial tissue compartment of these mice but did not develop tumors. Nevertheless, B-cell lymphomas developed in various organs, such as lung, liver, pancreas, and spleen, adjacent to VAV1-expressing epithelial cells. RAC1-GTP levels were unchanged in tissues from Rosa26 VAV1 mice, whereas ERK phosphorylation was increased in lymphomas [82]. Epithelial cells expressing VAV1 secrete CSF-1 and possibly other cytokines that influence the development of B-cell lymphoma [82]. This is the only VAV1 GEMM to show tumor development without additional molecular lesions, suggesting that VAV1 alone may lead to tumor development under certain circumstances. Additional experiments are needed to prove this phenomenon.

VAV1 has been shown to be a prognostic marker for human cancer. It is therefore tempting to suggest that it could serve as a target for therapeutic agents. So far, the obvious VAV1 activity that suggested it as a drug target was its ability to function as a GEF for RAC1. Azathioprine, an agent that blocks VAV1 GEF activity, was shown to prevent metastasis in a mouse model of pancreatic cancer by inhibiting VAV1 function [112]. Moreover, in the PDAC-VAV1-GEMM model, we have previously shown that treatment with azathioprine significantly inhibits tumorigenesis by blocking VAV1 function [81]. However, if in the future it is proposed to use azathioprine to treat VAV1-positive cancer, it would be useful to determine whether VAV1 in the tumor leads to the activation of the RAC1 pathway before treatment. 

## Figures and Tables

**Figure 1 cells-12-01276-f001:**
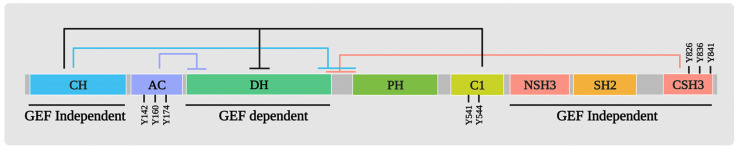
Structural domains of VAV1, their effector activities and control of its GEF activity. VAV1 protein contains the following domains: CH, calponin homology; Ac, acidic region; DH, Dbl homology; PH, pleckstrin homology; C1, a zinc like finger; NSH3, N-terminal SRC homology (SH3) domain; SH2, SRC homology 2; and CSH3, C-terminal SH3. Whether a region functions in GEF-dependent or -independent pathways is indicated beneath the region and is also detailed in the text. The intricate control of VAV1′s DH domain’s GEF activity by other domains of VAV1 is marked above the figure and explained in the text. Moreover, tyrosine residues that play a role in the control of VAV1 activity are marked. This figure was created with the BioRender.com program.

**Figure 2 cells-12-01276-f002:**
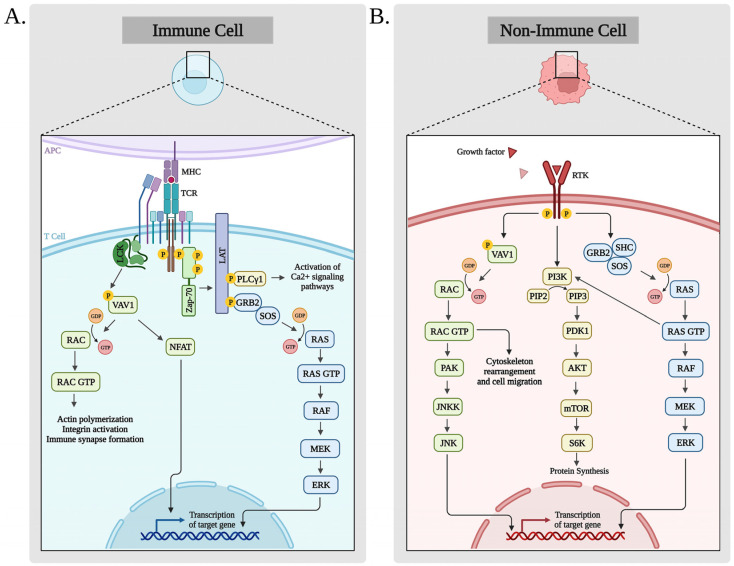
Activation of VAV1 tyrosine phosphorylation following T cell Receptor ligation (**A**) and EGFR activation (**B**). The figure illustrates the activation of VAV1 tyrosine phosphorylation in response to two distinct stimuli: T cell receptor (TCR) ligation (**A**) and epidermal growth factor receptor (EGFR) activation (**B**). (**A**): Engagement of the TCR with the MHC antigenic peptide complex, which recruits the cytoplasmic tyrosine kinase LCK and in turn activates VAV1 tyrosine phosphorylation. (**B**): EGF receptor activation initiates a diverse array of cellular pathways via dimerization. Each receptor dimer recruits different SH2-containing effector proteins, triggering distinct signaling pathways and culminating in cellular responses. VAV1 is tyrosine phosphorylated by EGFR, once it is recruited to the phosphorylated receptor via its SH2 domain. This figure was created with the BioRender.com program.

**Figure 4 cells-12-01276-f004:**
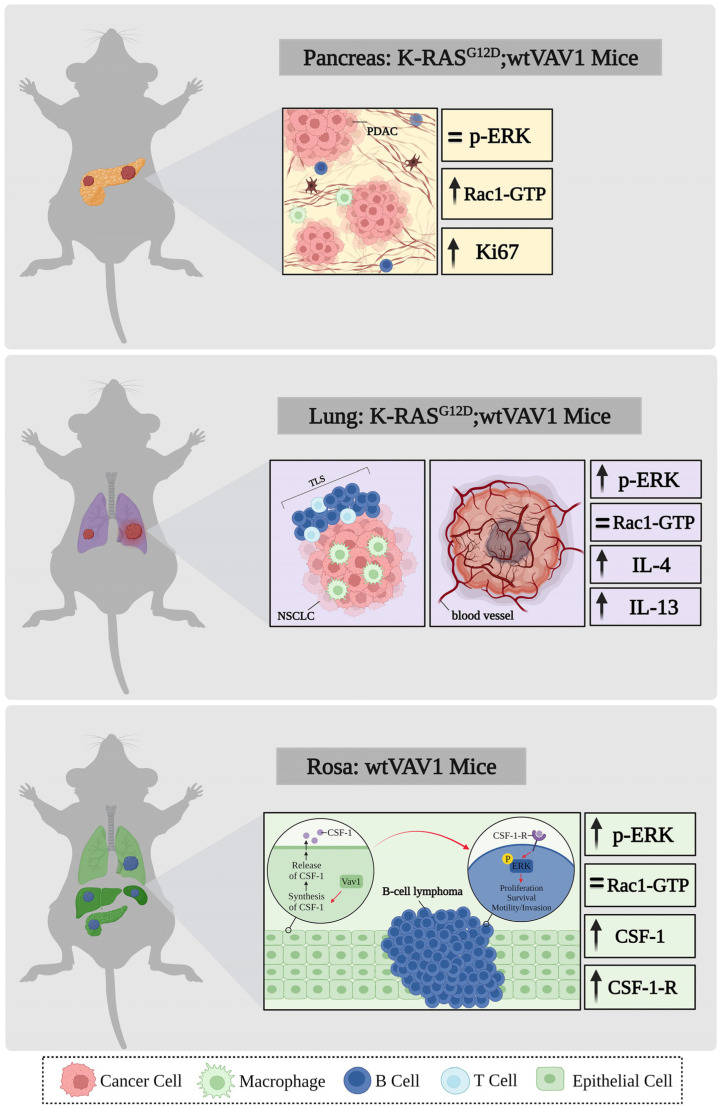
Development of tumors in GEMMs expressing wild-type (wt.) VAV1 in various organs. Results of cancer development and signaling pathways in GEMMs expressing wtVAV1 either in the pancreas [81], lungs [83], or throughout various organs [82], crossed with mice expressing mutant KRAS (pancreas, lungs) or alone (Rosa: Vav1 mice) are depicted. The figure illustrates the status of RAC1 and ERK activities in each mouse model, in addition to changes in cytokine expression. This figure was created with the BioRender.com program.

**Figure 5 cells-12-01276-f005:**
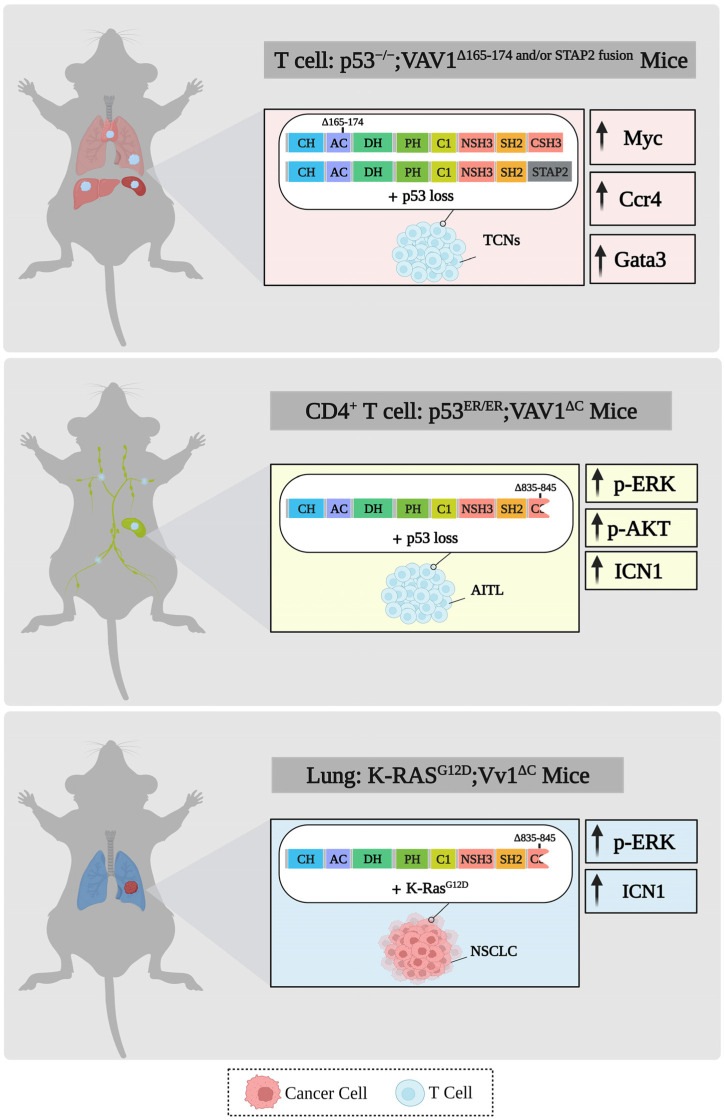
Development of tumors in GEMMs expressing mutant Vav1 genes in various organs. Results of cancer development and signaling pathways in GEMMs expressing Vav1^Δ165−174 and/or STAP2 fusion^ and p53 deletion in the hematopoietic system are depicted [110]. Moreover, the results of GEMMs expressing Vav1^ΔC^ and lacking p53 CD4^+^ T cells are demonstrated [111]. GEMMs that express Vav1^ΔC^ together with mutant KRAS in the lungs are also described [111]. The various tumors developed in the various GEMMs are indicated. This figure was created with the BioRender.com program.

## Data Availability

No new data were created or analyzed in this study.

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
