# Peer review of "Unraveling the Oncogenic Potential of VAV1 in Human Cancer: Lessons from Mouse Models"

_cells, 2023, doi:10.3390/cells12091276_

Round 1

Reviewer 1 Report

This review describes the role of VAV1 in human cancer.  It describes the structure and function of VAV1 and its expression and mutation in human cancers. The authors further dissect the literature to demonstrate differences in VAV1 overexpression and mutation depending on the type and location of the malignancies. Furthermore, the authors detail the possible mechanism of the functional effects of mutation in inducing cancer and conclude with a discussion of the potential to pharmacologically target VAV1.

This is a very well-written review that comprehensively and faithfully covers the literature available on the topic. The review was logically structured, and clearly written, making it very enjoyable to read. I recommend publication in its current form.

Author Response

Thanks to reviewer 1 for his review. Since there were no specific  requests for changes, we have no response.

Reviewer 2 Report

Review of Shalom et al.

1) Please use gene (italics) and protein (non-italics) definitions where appropriate.

2) Carefully define the difference between oncogene (VAV1 in italics) and oncoprotein (VAV1) in text.

3) Occasional mistakes, spelling errors, i.e. VAV instead of VAV1 (Intro), (Sos)1 – should be Sos1, etc.

4) Please check paragraph structure. 1-2 sentences do not make a paragraph, need to have a more substantial number of sentences and a clear point being made within a specific paragraph.

5) Figure 1 – please highlight potential tyrosine phosphorylation (and other PTM sites) on the line diagram.

6) Please use single amino acid codes to define key sites e.g. Y174, and phosphorylation should be pY174. To fully denote non-phospho and phospho forms, VAV1-Y174 and VAV1-pY174, etc.

7) The section on growth factor and RTK signaling in regulating VAV1 phosphorylation is poor. Please reformat and add a figure to explain how upstream signaling  causes VAV1 tyrosine phosphorylation and thus activation. What is missing is a clear mechanistic scheme depicting whether it is the RTK or a soluble tyrosine kinase e.g. c-Abl, c-Src or FAK that is directly targeting VAV1, especially in non-immune cells which may lack immune signaling capacity linked to TCR activation.

8) The section and discussion of VAV1 mutations in cancer is confusing. This needs to be rephrased in terms of gain-of-function and loss-of-function. If VAV1 is indeed an oncoprotein, somatic or germline mutations should confer additional or new activities for gain-of-function. In gene truncations or translational frameshifts, these are likely to be loss-of-function: so how can this be reconciled with cancer-promoting activity in VAV1?

9) The Fig. 3 schematic should have an additional panel with perhaps a view of the VAV1 molecule and where key mutations are located. There are a number of VAV1 structures in PDB which could be annotated to show the key residues that are mutated and how this could be reconciled with protein-protein interactions and signaling events.

11) The numbering of the sections is complicated for a review of this type and needs some streamlining.
